# OpenReview forum: "Convergence of Adam in Deep ReLU Networks via Directional Complexity and Kakeya Bounds"
_ICLR.cc/2026/Conference — ICLR 2026 Conference Withdrawn Submission_

### Official Review · Reviewer_6HgV · 2025-10-27

**Soundness:** 2
**Presentation:** 1
**Contribution:** 1
**Rating:** 2
**Confidence:** 3

**Summary:**

This paper studies the convergence property of Adam on deep ReLU neural networks trained over the Mean-Squared-Error loss. The paper pursues the analysis from the perspective of the regions separated by the ReLU activation. In particular, the paper argues that Adam only traverses polynomially many regions, and in each region, the loss function is effectively L-smooth. By assuming the boundedness of the layer norms, margin-based cut-off, spectral floor of Adam's second moment, low-rank gradient and sparsity, sparse Tope bound, sub-Gaussian drift control, angular concentration, directional richness, and the uniform low-barrier property, the paper shows that Adam goes through a two-phase convergence, where in the first phase the norm of the gradient converges sub-linearly, and in the second phase the loss and the distance to the global minimum converges linearly. The paper also provides generalization bounds of the learned solution by using the Kakeya-based covering number bound.

**Strengths:**

1. Studying the convergence of optimization algorithms on deep ReLU neural networks is a challenging and meaning task
2. The paper performs the analysis from the novel perspective of the regions separated by the ReLU decision boundary, and utilizes advanced mathematical tools such as Kakeya bound and Whitney fans.

**Weaknesses:**

1. The paper makes too many assumptions for the analysis, with most of them being not realistic. For instance, the paper requires a bounded norm of the layer weights and the stochastic gradient throughout training. This is something that should be established theoretically in a convergence proof. Moreover, the paper assumes L1-L7 together with an additional assumptions of the uniform low-barrier property given in the appendix. Most of these assumptions is precisely the technical difficulty in the analysis of the convergence property of training neural networks, and cannot be easily verified in the general case.
2. The paper tackles the convergence of Adam, but gives no novel perspective from the theoretical result of why we should apply Adam in training neural networks rather than the simple SGD.
3. The paper leaves out important settings, notations, and definitions undefined. For instance, the Adam algorithm is not presented in the paper, thus making it unclear what is $\hat{v}_t$ in L2, and what is $\beta_1, \beta_2$ in Theorem 3. The "mask-freeze" property assumed on page 7 should be L1, but not explicitly stated. It is also not clear what the post-freeze iterates in the statement of Theorem 4 refers to.
4. The paper has no experimental results that validates its theoretical results. The property that Adam only traverses polynomially many regions should be an easy to study property, and is a central argument of the paper. However, there is no experiments that validates this claim.

**Questions:**

None

---

### Official Review · Reviewer_xDP6 · 2025-10-28

**Soundness:** 1
**Presentation:** 1
**Contribution:** 1
**Rating:** 0
**Confidence:** 4

**Summary:**

This paper claims to obtain (i) generalization bounds with better dimension dependence compared to existing results and (ii) the first global optimization results for ReLU networks trained with Adam.

**Strengths:**

Original attempt at analyzing the generalization and optimization of ReLU networks trained with Adam.
The idea is indeed novel and has the potential to bring about important results.

**Weaknesses:**

### 1. Absence of  references for technical results used

This paper leverages many very technical existing results. However, they are almost always mentionned without appropriate refernces!
A few examples:
- l679: "Goresky-MacPherson"
- l692: "Whitney regularity implies..."
- l816: "Zaslavsky's theorem..."
- l846: "Stratified Morse theory gives the bound..."
- l854: "Smith theory implies..."
- l1782: "Applying Freedman's inequality..."
- l1882: "Standard Adam descent lemma"

Moreover, a few references are mentionned but not included in the bibliography (eg l827).

### 2. Missing key elements of proofs
- l826: what is $h(\theta)$ and what is its use?
- What are $\Pi_{cones}$ and $\Pi_{crossings}$? Even afcter reading App. B, I still do not understand.
- "Assumption A.1 in Appendix A" l376  in the main text does not exist.
- Wang and Zahl, 2025 (ref. l564 in the bibliography) only provides results in dimension 3 while l1917 claims it provides a result in arbitrary dimension $d$. Moreover, the reference to Wang and Zahl 2025 l2004 is hallucinated.
- Lemma 19: the assertion "A low barrier path $\gamma$ connects..." l2078 does not follow from Assumption 7.
- "L7 - Directional Richness" l230 does not seem to be formally stated.
- The "Consequently..." part of Theorem 3 is not

### 3. Missing main claim
One of the missing main claims: the UBL property (l044-050), which is presented in the intro as a main claim, does not seem to be stated nor  proven anywhere.



### 4.Assumptions not coherent with the setting
- In D.2, "Key Assumptions" include smoothness of the loss in $\theta$, which contradicts the contributions (eg l64)

### 5. Bounded  gradients

Assumption A.1 include boundedness of the weights: how is this possible in general? (also appears in assumptions  in the appendix) This should be discussed.


### 6. Minor
- Thm. 6 and Thm. 9 are the same.
- Formatting issues in E.1
- References at l2002 and l2065 should be added to the main bibliography


### 7. Missing references
 The literature on Adam is much larger. The literature on generalization bounds andin particular PAC-Bayes bounds that take into account the geometry of the algorithm is not mentioned.

**Questions:**

- Could you detail Assumption 4 in D.3? And explain how it is used in Lemma 7?
- Could address Weaknesses 2-5?

---

### Official Review · Reviewer_adrx · 2025-10-29

**Soundness:** 1
**Presentation:** 1
**Contribution:** 3
**Rating:** 2
**Confidence:** 3

**Summary:**

The submitted manuscript analyzes Adam for training of deep ReLU networks without assuming global smoothness or convexity. The analysis is based on stratified Morse theory and Kakeya geometry. The authors claim to derive the first $\tilde O(\sqrt{d_{eff}/n})$ generalization bound for Adam and a global convergence result in the non-smooth ReLU landscape via a two-phase rate: sub-linear phase followed by exponential decay towards a minimizer. The non-smoothness is handled using a finite region-crossing complexity, whereas the Kakeya covering argument yields a generalization result.

**Strengths:**

- The paper tackles a challenging and timely theoretical question about the convergence and generalization behavior of Adam in non-smooth deep learning settings.
- The technical development is ambitious, combining ideas from geometry, Morse theory, and optimization in an original way.
- The idea of handling non-smoothness through hyperplane-crossing analysis is neat and conceptually appealing.
- I believe the work could be of large interest to the optimization theory and deep learning theory communities, given its attempt to unify geometric and probabilistic perspectives.

**Weaknesses:**

While I believe the paper has the potential to be a strong contribution to our understanding of the Adam optimizer’s behavior, I have several concerns regarding the writing, rigor, and correctness of the work.
- I understand that the formulations need to be written densely to fit within the page limit. However, in several places, intermediate explanatory text is missing, which interrupts the reading flow and makes the paper difficult to follow. For example, there is no contextual text preceding Theorem 1, Theorem 3, or Theorem 4, which leaves these main results introduced abruptly.
- The paper is written in a very unclear way. Many important definitions are missing or appear only deep in the 42-page appendix, without any reference in the main body. In particular, the main algorithm under study, Adam, is never defined in the main text nor properly referenced to the appendix. Even in the appendix, the dynamics are scattered across different proofs (e.g., Lemma 8, Corollary 2) with missing details. For instance, the momentum term $\hat m_t$ is never explicitly defined.
- Numerous assumptions are introduced throughout the paper, yet they are not clearly stated alongside the theoretical results. Some appear only within the proofs. Consequently, several results are described incompletely, and the overall presentation quality falls short of the standards expected for a theoretical paper.
- Due to these issues, it is often difficult to interpret the main results. For example, it is unclear in what sense the error bounds are stated — whether they hold in high probability, almost surely, or in expectation. Since the Adam algorithm defines a stochastic process $\theta_t$, such distinctions are essential. The results in Theorem 3 are presented as if deterministic, though they should presumably hold in expectation or with high probability. Moreover, the proof of this theorem appears overly short and leaves major logical gaps — for instance, the so-called “Adam descent lemma” (line 1883) is neither justified nor clearly stated. Similarly, Theorem 5 is written as a deterministic convergence claim, but it remains unclear whether convergence is to a specific or arbitrary optimum $\theta^*$?
- The submitted paper is very long and includes many proofs, yet several appear informal, incomplete, or incorrect. For example, the proof of Lemma 11 (again, with no assumptions stated in its formulation) does not make sense as written. In many places, big-O notation is used inconsistently, as if it represented non-asymptotic quantities (see, e.g., the proofs of Lemma 14 and Theorem 15, line 1885). Further examples of unclear or incorrect reasoning are mentioned in my questions below.

**Questions:**

- In Theorem 3, how is the convergence of the last iterate of the gradient norm established? The argument as written does not clearly imply convergence of the last iterate norm.
- What are the precise assumptions required for the main results (Theorems 1–5)? Please specify them clearly instead of referring implicitly to scattered lemmas or appendix material.
- The paper assumes both the Polyak–Łojasiewicz (PL) condition and the Kurdyka–Łojasiewicz (KL) property. Since the PL condition is typically a special case of the KL property, why are both needed? Moreover, the PL condition is only defined in the proof of Theorem 7 in the appendix, while KL is never defined explicitly, making it difficult to understand what version of the property is used.
- The proof of Theorem 13 invokes the assumption $N\gg d_{eff}$, but this assumption is not stated anywhere. Why should this imply the final asymptotic bound $O(\log(N) d_{off})?
- In Assumption 5, what is the intended logical quantification? Does there exist a single $\delta$ satisfying the stated conditions, or should the conditions hold for any $\delta$. If it is the latter, the choice $\delta = O(1/T)$ in Lemma 14 seems infeasible. Since Lemma 14 is used in the proof of Theorem 14, this apparent inconsistency should be clarified.
- In Lemma 13, why should $d_t$ be assumed to be uniformly lower bounded in $t$? This is not mentioned as an explicit assumption and it appears to be very restrictive.
- How is equation (15) derived from (12)-(14)? How do you obtain (16) from (15)? If $\delta_t$ is of order $1/t$ this statement appears to be wrong.
- Why is it feasible to assume a lower bound on $v_t$ in Theorem 7?
- Are the conditions L1–L7 assumed throughout the paper, or only within specific results?
- In line 280, why should the quantities $B$ and $GR$ be finite almost surely?

Typos and inconsistencies:
- Heading of Section 4: Emp[i]rically (also occurs in line 192)
- Abstract (Line 18-19): non[-]smooth, non[-]convex
- Abstract (Line 19): relu -> ReLU
- Line 174: [s]tratified
- Line 298: hyper-plane -> hyperplane
- Inconsistency between sub-Gaussian and subgaussian.
- Various abbreviations are introduced multiple times.
- Line 503: Authors listed twice.
- Line 508-510: Reference listed twice.
- Line 569: Title listed twice.
- In Theorem 2 $\Pi_{cones}$ should be $\Pi_{crossings}$?
- Many points and commas are missing in listings (e.g. lines 1651-1655) or equations (throughout the document).
- N_{crossings} is undefined. Is it the same as $\Pi_{crossings}$?

---

### Official Review · Reviewer_1LTB · 2025-10-31

**Soundness:** 3
**Presentation:** 3
**Contribution:** 2
**Rating:** 4
**Confidence:** 2

**Summary:**

This paper derives generalization bound for Adam in deep ReLU networks and establishes the global convergence guarantees for Adam in the non-smooth, non-convex ReLU landscape without relying on global Polyak- Lojasiewicz (PL) or convexity assumptions. The
core of the analysis is a multi-layer refinement framework based on stratified Morse theory and results in Kakeya sets. This framework leads to a  generalization bound than those derived from PAC-Bayes approaches, marking a  theoretical advance.

**Strengths:**

-  The paper provides the global convergence theorem for Adam in deep ReLU networks under non-smooth, non-convex conditions, without NTK linearization or convexity assumptions.

- The paper presents a framework combining stratified Morse theory and Kakeya sets, which offers a fresh perspective on optimization dynamics.

- By leveraging assumptions (L1-L7), which are motivated by empirical observations, this paper reduces region-crossing complexity from $O(N^d)$ to $O(d_{\mathrm{eff}}\log N)$, consequently deriving stronger convergence guarantees.

**Weaknesses:**

- The derived result is quite abstract and dense. The manuscript would benefit greatly from more intuitive explanations, perhaps accompanied by simple diagrams or illustrative examples, to help the reader build a conceptual understanding of the core geometric arguments.

- While the paper mentions that assumptions (L1-L7) are ``motivated by empirical findings" and provides brief citations. it lacks detailed, convincing explanations or specific empirical results to support these claims.

- The empirically motivated assumptions L1-L7 may be a bit strong. The paper dose not discuss cases where these assumptions fail.

**Questions:**

- There are several typos in the manuscript (e.g., "emperical" should be "empirical", which occurs multiple times).

- The paper introduces a large number of mathematical symbols. Adding a table of notation could enhance readability and accessibility.

- The write up could be improved.

---

### Note · Authors · 2025-11-25

I have read and agree with the venue's withdrawal policy on behalf of myself and my co-authors.